# Exploring the Correlation between Likelihood of Flow-based Generative Models and Image Semantics

## Abstract

Among deep generative models, flow-based models, simply referred as *flow*s in this paper, differ from other models in that they provide tractable likelihood. Besides being an evaluation metric of synthesized data, flows are supposed to be robust against out-of-distribution (OoD) inputs since they do not discard any information of the inputs. However, it has been observed that flows trained on FashionMNIST assign higher likelihoods to OoD samples from MNIST. This counter-intuitive observation raises the concern about the robustness of flows' likelihood. In this paper, we explore the correlation between flows' likelihood and image semantics. We choose two typical flows as the target models: Glow, based on coupling transformations, and pixelCNN, based on autoregressive transformations. Our experiments reveal surprisingly weak correlation between flows' likelihoods and image semantics: the predictive likelihoods of flows can be heavily affected by trivial transformations that keep the image semantics unchanged, which we call semantic-invariant transformations (SITs). We explore three SITs (all small pixel-level modifications): image pixel translation, random noise perturbation, latent factors zeroing (limited to flows using multi-scale architecture, e.g. Glow). These findings, though counter-intuitive, resonate with the fact that the predictive likelihood of a flow is the joint probability of all the image pixels. So flows' likelihoods, modeling on pixel-level intensities, is not able to indicate the existence likelihood of the high-level image semantics. We call for attention that it may be *abuse* if we use the predictive likelihoods of flows for OoD samples detection.

## 1 Introduction

Deep generative models have been very successful in image generation (Brock et al., 2018; Kingma & Dhariwal, 2018; Miyato et al., 2018), natural language generation (Bowman et al., 2015; Yu et al., 2017), audio synthesis(Van Den Oord et al., 2016) and so on. Among them, generative adversarial networks (GANs) are implicit generative models(Goodfellow et al., 2014) that explicit likelihood function is not required, and are trained by playing a minimax game between the discriminator and the generator; Variational auto-encoders (VAEs,Kingma & Welling (2013); Rezende et al. (2014)) are latent variable generative models optimized by maximizing a lower bound, called evidence lower bound, of the data log-likelihood. Flow-based models (Dinh et al., 2016; 2014; van den Oord et al., 2016) differ from them in that they provide exact log-likelihood evaluation with *change of variables* theorem (Rezende & Mohamed, 2015). A flow usually starts with a simple base probability distribution, e.g. diagonal Gaussian, then follows a chain of transformations in order to approximate complex distributions. Each transformation is parameterized by specially designed neural networks so that the log-determinant of its Jacobian can be efficiently computed.

Most of the previous works focus on how to design more flexible transformations to achieve tighter log-likelihoods, and generate more realistic samples. It is also believed that flows can be used to detect out-of-distribution(OoD) samples by assigning low likelihoods on them. However, it has been observed that flows fail to do so. For example, flows trained on FashionMNIST surprisingly assign higher likelihoods on MNIST samples (Nalisnick et al., 2018; Choi & Jang, 2018). Though analyses on pixel-level statistics are performed on this phenomenon (Nalisnick et al., 2018), and

density evaluation combined with uncertainty estimation is used to detect OoD samples (Choi & Jang, 2018), the reasons behind flows' counter-intuitive behaviours are still not clear.

Humans easily discriminate MNIST images from FashionMNIST images, since their high-level image semantics are perceptually different. Accordingly, it takes some metrics that can reflect the high-level image semantics for OoD detection. In this paper, we empirically explore the correlation between flows' likelihoods and image semantics, and question the rationality and applicability of using predictive likelihoods of flows for OoD detection. We first introduce a concept of semantic-invariant transformation (SIT). An SIT transforms an input *without* changing its high-level semantics, e.g. a dog image through an SIT is still supposed to be recognized as a dog. We choose two typical flow-based models as target models: Glow (Kingma & Dhariwal, 2018), based on coupling transformations, and pixelCNN (van den Oord et al., 2016), based on autoregressive transformations. We evaluate on image datasets MNIST and FashionMNIST under three trivial SITs: image translation, random noise perturbation, and latent factors zeroing (specific to invertible flows using multi-scale architectures, e.g. Glow).

We demonstrate that the predictive likelihoods of the target models show weak correlation to the image semantics in the following ways:

- Small pixel translations of test images could result in obvious likelihood decreases of Glow.
- Perturbing small *random* noises, unnoticeable to humans, to test images could lead to catastrophic likelihood decreases of target models. This also applies even if we keep the semantic object of a test image intact, and only add noises to the background.
- For an invertible flow using multi-scale architecture, e.g. Glow, the inferred latent variables of an image is a list of gaussianized and standardized factors. We find that the contributions of a flow's blocks to the log-likelihood are constant and independent of inputs. Thus, simply zeroing the preceding latent factors of a sample image, and feed them to flow's reverse function. We could obtain new samples with surprisingly higher likelihoods, yet with perceptually unnoticeable changes from the original image.

We emphasize that all these SITs are small pixel-level modifications on test images, and undoubtedly have no influences on humans' recognition of the semantic objects in the images. However, they lead to obvious inconsistency of flows' likelihoods on test samples. Considering that the predictive likelihood of a flow is the joint probability of all the image pixels, it may not convincingly indicate the existence of a semantic object in an image. Thus it could be problematic to use flows for downstream tasks which require metrics that can reflect image semantics, e.g. OoD detection.

## 2 BACKGROUND

### 2.1 CHANGE OF VARIABLES THEOREM

Given a random variable $z$ with probability density function $p(z)$, after applying an invertible function $f : \mathcal{R}^D \to \mathcal{R}^D$ on $z$, we get a new random variable $z' = f(z)$. Then probability density function of the changed variable $z'$ is given by:

$$p(z') = p(z) \big| \det \frac{\partial f^{-1}}{\partial z} \big| \tag{1}$$

We can construct arbitrarily complex probability distributions by transforming a simple base distribution $p(z_0)$ with a chain of mappings $f_k$ of length $K$. Then we have :

$$\log p(z_K) = \log p(z_0) + \sum_{k=1}^{K} \log \big| \det \frac{\partial f_k^{-1}}{\partial z_{k-1}} \big| \tag{2}$$

### 2.2 FLOW-BASED MODELS

Flow-based models are generative models designed by applying the above theorem, thus exact log-likelihood evaluation of data is feasible. Then the practical problem of building flow-based models

applied on high-dimensional data, like images, becomes how to design invertible transformations whose Jacobian determinant can be efficiently computed.

Research on flows is very active and rapidly evolving. In this paper, we particularly focus on flow-based generative models on images and the behaviours of their likelihoods. They can roughly be divided into two categories according to the granularity of the transformation layers:

**Coupling Flow**    A coupling flow contains a sequence of coupling layers which model the transformation in a coarse way. Coupling layers split a $D$-dimensional intermediate random variable $\boldsymbol{z}$ into two parts: $\boldsymbol{z}_{1:d}, \boldsymbol{z}_{d:D}$. A general form of coupling layer is affine(Eq. 3). The first part $z_{1:d}$ is kept still, while the second part $\boldsymbol{x}_{d:D}$ is scaled and shifted with transformations $s, t$ on the first part $\boldsymbol{x}_{1:d}$.

$$
\begin{aligned}
\boldsymbol{y}_{1:d} &= \boldsymbol{x}_{1:d}, \\
\boldsymbol{y}_{d:D} &= \boldsymbol{x}_{d:D} \cdot \exp(s(\boldsymbol{x}_{1:d})) + t(\boldsymbol{x}_{1:d}).
\end{aligned}
\tag{3}
$$

The affine coupling is proposed in Real NVP (Dinh et al., 2016), whose Jacobian is a lower triangular matrix that can be efficiently computed. An earlier and simpler version is additive coupling proposed in NICE (Dinh et al., 2014), which can be obtained by simply removing the scale item $\exp s(\boldsymbol{x}_{1:d})$ in affine coupling. Additive coupling layer is volume-preserving and the log-determinant of its Jacobian is always 0. Glow improves Real NVP by replacing the fixed shuffling permutation with $1 \times 1$ invertible convolution. Since forward and inverse operation of a coupling layer have the same computational efficiency, both likelihood evaluation and sampling(or generation) for coupling flows are equally efficient.

**Autoregressive Flow**    As the building blocks of autoregressive flow, autoregressive transformations model the joint probability $p(\boldsymbol{x})$ as the product of one-dimensional conditionals:

$$
p(\boldsymbol{x}) = \prod_{i=1}^{D} p(\boldsymbol{x}_i | \boldsymbol{x}_{<i}),
\tag{4}
$$

where the probability of observation $\boldsymbol{x}_i$ conditions only on its previous observations $\boldsymbol{x}_{<i}$. The autoregressive property of an autoregressive layer is enforced by specially designed mechanism, e.g. masking.

In PixelCNN (van den Oord et al., 2016), this is implemented as masked convolutional layers, which are inherently easier to be parallelized than its counterpart PixelRNN (Oord et al., 2016). The likelihood evaluation of PixelCNN takes only one-forward pass, but its inference, i.e. generation, takes $O(D)$, since we have to sample pixel-by-pixel. PixelCNN can be further parallelized (Reed et al., 2017b) to accelerate its inference speed.

## 3    OTHER RELATED WORKS

**Flows for generation**    Serving as powerful decoders, PixelCNN can also be combined with other generative models, e.g. PixelGAN (Makhzani & Frey, 2017) and PixelVAE (Gulrajani et al., 2016). Variants of PixelCNN are also used to model audio (Van Den Oord et al., 2016), video (Kalchbrenner et al., 2017), and text (Kalchbrenner et al., 2016). PixelCNN, combining with attention, is also applied to few-shot autoregressive density estimation Reed et al. (2017a). Ho et al. (2019) proposes several improvements to coupling flow, reducing its gap to autoregressive flow in terms of density estimation.

**Autoregressive models for density estimation**    Autoregressive models can be specially designed for general-purpose density estimation. Masked Autoencoder for Distribution Estimation (MADE, Germain et al. (2015)) is a pioneering work that use masked neural networks to model the autoregressive density. MADE constitutes the building block of two popular *normalizing flows*: Inverse Autoregressive Flow (IAF, Kingma et al. (2016)) and Masked Autoregressive Flow (MAF, Papamakarios et al. (2017)). IAF and MAF are similar but with different computational trade-offs. IAF, providing efficient sampling, is designed to improve the expressiveness of the approximate posterior of VAE. MAF is a more powerful density estimator which stacks multiple MADEs.

# 4 METHOD

Let $g$ be a flow-based generative model trained on dataset $X = \{x_i\}_{i=1}^N$ sampled from some unknown $p(x)$, and $p_g(x)$ be the $g$'s predictive probability density function of sample $x$ .

**Semantic-Invariant Transformation (SIT)**   Roughly speaking, SIT can be any transformation that do not change humans' recognition of image semantics. For example, suppose $x$ is a dog image. After applying SIT $T$ to $x$, $T(x)$ is supposed to be high recognized as a dog image. As a proof of concept evaluation, we limit our evaluations to three trivial SITs: image translation, random noise perturbation, and latent factors zeroing (specific to Glow).

**Semantic Correlation**   We probe the correlation between the predictive likelihood of a flow $g$ and image semantics by examining the influences of SITs on test samples' likelihoods. Specifically, a reasonable observation we should expect is that for SIT $T$: $\left| p_g(T(x)) - p_g(x) \right| < \delta$ holds for a small positive scalar $\delta$.

We report *bits-per-dim* (BPD), which is given by:

$$\text{BPD} = \frac{\text{NLL}}{(h \times w \times c) \cdot \log 2} \tag{5}$$

where NLL is the negative log-likelihood of the test sample, $h, w, c$ are height, width, number of channels. Lower BPD implies higher likelihood. Throughout this paper, we use BPD and likelihood interchangeably. We refer to Supp. A for setup and training details of the target models.

## 4.1 IMAGE TRANSLATION

Translation invariance is a fundamental property in learning image representations that are robust for downstream tasks. In this section, we evaluate the influences of image translations on flows' likelihoods.

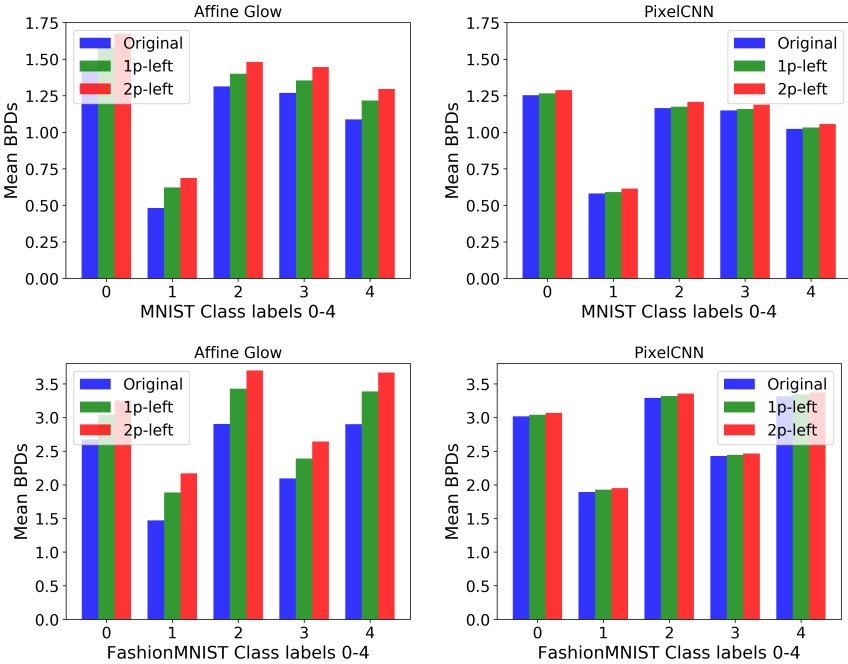

Figure 1: BPDs of (affine) Glow and PixelCNN on MNIST test set with 1-pixel and 2-pixel left translation. Shown are class labels from 0 to 4. See Supp. B.1 for additional results.

The results in Fig. 1 and examples in Fig. 2 show that even 1 or 2-pixel left translation could lead to obvious increase of Glow's predictive BPDs, while the PixelCNN's predictive BPDs are robust to

pixel translation. This surprising difference can be attributed to the difference of their architectures. Glow, like other flows based on coupling transformation layers (Dinh et al., 2016), models the joint probability of the pixels in a coarse-grained way. They rely on multi-scale architecture modeling different level of abstractions in order to achieve competitive BPDs. At higher scale levels, the intermediate tensors have smaller spatial sizes and bigger channel sizes. This is performed at the starting point of each scale level with **squeeze** operation, which trades spatial sizes for channel sizes by transforming a $h \times w \times c$ tensor into a $h/2 \times w/2 \times 4c$ tensor. Note that **squeeze** operation actually *destroys* the spatial positions of adjacent pixels, and 1-pixel translation could lead to quite different spatial partitions. While for PixelCNN, the intermediate tensors are not reshaped, and the spatial positions of pixels are kept still; Furthermore, the prediction of each pixel conditions only on a neighborhood of (previous) pixels in masked convolution, so translation invariance is preserved.

| | original | 1p-left | 2p-left | | original | 1p-left | 2p-left |
|---|---|---|---|---|---|---|---|
| Affine | 0.852 | 1.026 | 1.118 | | 1.715 | 1.838 | 2.083 |
| Additive | 0.981 | 1.066 | 1.132 | | 1.761 | 1.866 | 2.007 |
| PixelCNN | 0.851 | 0.849 | 0.859 | | 1.948 | 2.030 | 2.088 |
| Affine | 1.306 | 1.440 | 1.461 | | 2.967 | 3.652 | 3.842 |
| Additive | 1.483 | 1.656 | 1.701 | | 3.217 | 3.494 | 4.252 |
| PixelCNN | 1.099 | 1.104 | 1.114 | | 2.780 | 2.810 | 2.859 |

Figure 2: Examples from MNIST, FashionMNIST test sets with 1-pixel and 2-pixel left translations. Below are predictive BPDs of Glows (Affine and Additive) and PixelCNN.

**Problematic Likelihood Comparisons**    The foundation of using likelihood-based models for OoD detection is that they are supposed to assign much lower likelihoods for OoD samples $\boldsymbol{x}_{out}$ than in-distribution samples $\boldsymbol{x}_{in}$, i.e. $p_g(\boldsymbol{x}_{in}) \gg p_g(\boldsymbol{x}_{out})$. However, it has been observed that flows assign higher likelihoods on OoD samples than even training samples (Nalisnick et al., 2018; Choi & Jang, 2018). Analyses of pixel-level datasets statistics in (Nalisnick et al., 2018) show that this may be due to OoD datasets just "sit inside of" in-distribution datasets with roughly the same mean, smaller variance. Surprisingly, similar counter-intuitive likelihood assignment also occurs in in-distribution samples. For example, in Fig. 1, images with class label 1 are consistently have significantly lower BPDs, i.e. higher likelihoods than samples of other classes. In OoD detection, we assume that a sample with a higher likelihood indicates that it is more likely to be an in-distribution sample. Following the same logic, this tells that all images of class label 1 are more likely to be in-distribution samples than samples from other classes, which contradicts the fact that they are all in-distribution samples for sure. We may reasonably suspect that flows' counter-intuitive likelihood assignment is dominated by the inherent differences of pixel-level statistics associated to the image semantics, e.g. different numbers. This kind of counter-intuitive likelihood comparisons exist not only between in-distribution and OoD samples, but also within in-distribution samples from different classes. Similarly, Theis et al. (2015) find that 1-pixel shift of the images could lead to quite different nearest neighbours from the training set, measured in Euclidean distance, which also demonstrate the gap between pixel-level metrics and humans' perception.

## 4.2   RANDOM NOISE PERTURBATION

Image pixels are discrete integers; and in practice, right amount of real-valued *uniform* noise is added to dequantize the pixels. For images $x \in [0, \dots, 255/256]^D$ scaled to $[0, 1]$, we usually do $\boldsymbol{x} = \boldsymbol{x} + \boldsymbol{u}, \boldsymbol{u} \in [0, 1/256]^D$. In our experiments, we find that adding small *random* perturbations out of the coverage of the added noise, i.e. $\boldsymbol{u} \in [0, 1/256]^D$ here, to test images cause flows to give catastrophically higher BPDs.

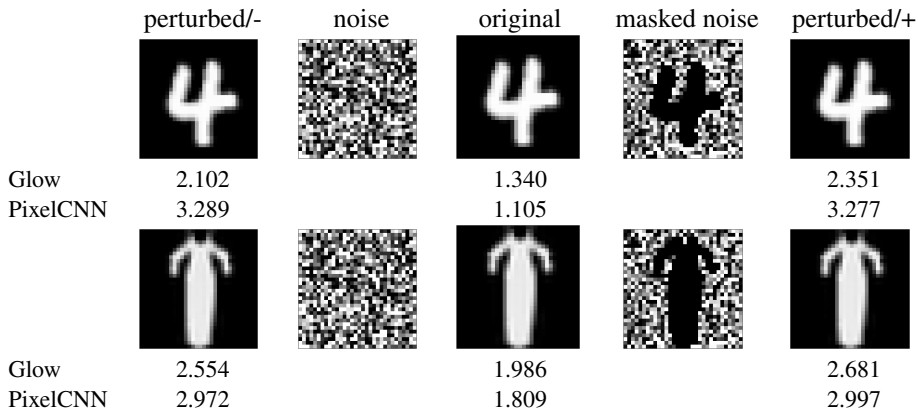

|  | perturbed/- | noise | original | masked noise | perturbed/+ |
|---|---|---|---|---|---|
| Glow | 2.102 |  | 1.340 |  | 2.351 |
| PixelCNN | 3.289 |  | 1.105 |  | 3.277 |
| Glow | 2.554 |  | 1.986 |  | 2.681 |
| PixelCNN | 2.972 |  | 1.809 |  | 2.997 |

Figure 3: Examples from MNIST, FashionMNIST test sets perturbed with gaissian noises. The middle column shows the original images. The second and fourth columns are the gaussian noises and masked gaussian noises. The first and fifth columns are the perturbed images with noises w/o (+/-) masks. Below the samples are the BPDs reported by Affine Glow and PixelCNN. Test images are scaled to $[0, 1]$. We could simply generate mask by setting a threshold for the scaled pixels. Here we use $\boldsymbol{m} = \boldsymbol{x} < 0.3$. See Supp. D for more examples.

Humans can robustly recognize semantic objects in images regardless of the backgrounds. So we also evaluate influences of adding random perturbations to only the backgrounds of test images. This can be simply implemented using proper mask:

$$\boldsymbol{x} \leftarrow \boldsymbol{x} + \epsilon \cdot \boldsymbol{m} \odot \text{noise} \tag{6}$$

where $\boldsymbol{m}$ is the mask, and $\epsilon$ is a small scaling factor ensuring the noise is small enough.

In our evaluations, we use unit Gaussian noises, and set the scaling factor $\epsilon = 0.001$. Examples in Fig. 3 manifest that adding small noises catastrophically lower samples' likelihoods. Compared to Glow, PixelCNN is more sensitive to the noises, because its pixel-wise modeling quickly augment and propagate the influences of the added noise. We get similar results even if we keep the semantic objects of test images intact, and add noises only to the backgrounds. Note that the Gaussian noises $\epsilon \cdot \mathcal{N}(0, 1)$ we added are out of the coverage (with $< 0$ elements) of the uniform noises $[0, 1/256]$ added during training, so theoretically this is expected since models are not optimized in that areas. However, it does reveal that flows are not aware of the image semantics, and treat the pixels of objects and the pixels of backgrounds with no discrimination. Other tested noises include $\epsilon \cdot (1/256 + \mathcal{N}(0, 1))$ and $\epsilon \cdot [-1/256, 0]$, and similar results were obtained.

### 4.3 FREE LIKELIHOOD LOOPHOLE OF GLOW

---
**Algorithm 1** Generate $\boldsymbol{x}^*$ by zeroing the latent factors

---
1: **Input**: image $\boldsymbol{x}$, label $y$ (*optional*), Glow model $g$, $k < L$.
2: $\boldsymbol{z} \leftarrow g.forward(\boldsymbol{x}, y)$          ▷ Infer the latent factors.
3: $\boldsymbol{z}^* \leftarrow zero(\boldsymbol{z}, k)$          ▷ Zero the preceding $k$ factors.
4: $\boldsymbol{x}^* \leftarrow g.reverse(\boldsymbol{z}^*, y)$          ▷ Reverse the zeroed latent factors.
5: **return** $\boldsymbol{x}^*$

---

Let us first decompose the Glow architecture into blocks and review their contributions to the final log-likelihood. A Glow consists of a sequence of modules at different scale levels. At each scale level, it starts with a *squeeze* operation, which reshapes the intermediate tensor without contribution to the log-likelihood. Following each *squeeze* operation is a stack of step flow blocks. A step flow block consists of three layers: actnorm, invertible $1 \times 1$ convolutional layer, and coupling layer. The log-determinants of actnorm and $1 \times 1$ convolutional layer are input-independent, and depend only on their inner weights (see Table 1 in (Kingma & Dhariwal, 2018)). Additive coupling layer is volume-preserving whose log-determinant is 0, thus is also input-independent. For affine coupling layer, its

log-determinant depends on the affined half, but is quantitatively small. Compared to Glow with additive coupling layers, affine layers bring only a small improvement of $< 0.05$ BPD (Kingma & Dhariwal, 2018). Then, the intermediary tensor is split into two halves along the channel dimension. One halve is gaussianized with a convolutional block, then the other halve is factored out after being standardized. This procedure significantly reduces the amount of computation and memory. We refer to Dinh et al. (2016) for more details due to limited space.

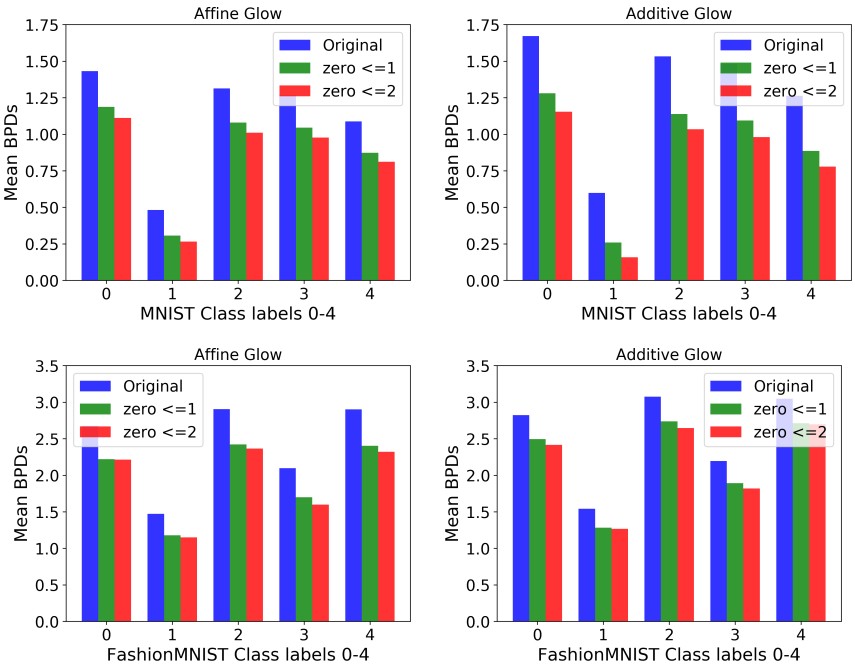

Figure 4: BPDs of affine and additive Glow on MNIST, FashionMNIST test sets by zeroing the preceding 1 (zero $z_{\leq 1}$) and 2 (zero $z_{\leq 2}$) latent factors. Shown are classes 0 to 4. See Supp. C for additional results.

So for a Glow using additive couplings, the cumulative log-determinant of the flow blocks *within* a particular scale-level is constant regardless of different samples. Only the log-determinants of gaussianized factorings (between the transitions of different scale-levels) depend on the individual inputs. Denote $z = \{z_1, \ldots, z_L\}$ as the latent variables of input image $x$. Each $z_i$ is the standardized vector of $i$-th scale level. This simply means that a sample $x$ whose latent variables $z$ are close to center $0$ will have a higher log-likelihood. Empirically, it also applies to Glow using affine coupling layers, since the influence of varying the latent variables on the log-determinants of affine coupling layers is quantitatively small compared to its gain.

We could make use of this property to generate samples with higher likelihoods via the invertibility of Glow for free (see Algorithm 1, label $y$ is optional). We find that the semantic object of a test image depends heavily on the last factored latent $z_L$, rather than the preceding factors. Examples in Fig. 5 show that zeroing the preceding 1 and 2 latent factors could give us samples with obviously lower BPDs but without obvious changes (slightly faded pixel intensities) of the semantic objects. Results in Fig. 4 show that zeroing the first latent factor give the maximum increment of likelihoods.

## 4.4 IMPLICATIONS ON DISCRIMINATIVE CLASSIFIERS

We also evaluate the influences of these SITs on the performance of discriminative classifiers. In contrast to the obvious change of flows' likelihoods, these small perturbations could decrease the testing accuracies of classifiers to some insignificant, or negligible (on MNIST) extent (see Tab 1). The discriminative classifier used here is a shallow residual network of 8 layers on $32 \times 32$ images in the structure specified in (He et al., 2016).

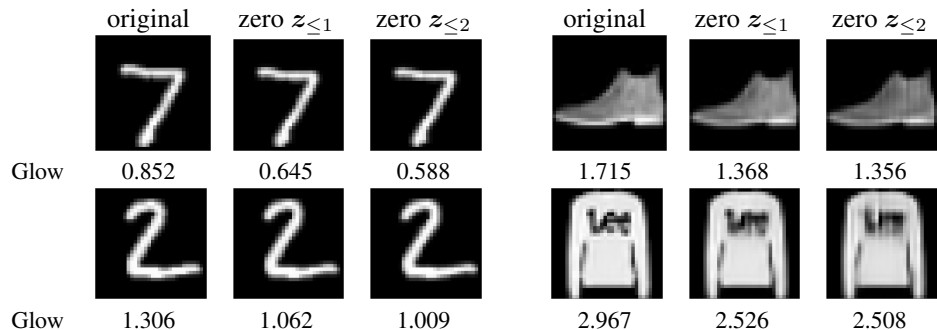

Figure 5: Examples of Affine Glow with different zeroed latent variables on MNIST (left) and FashionMNIST (right) test sets. See Supp. C.1 for results of Additive Glow.

| Perturbations | MNIST | FashionMNIST |
|---|---|---|
| Clean | 99.31% | 93.03% |
| 1-pixel left | 99.12% | 89.71% |
| Gaussian noises | 99.31% | 93.04% |
| Zero $<= 1$ | 99.18% | 91.44% |
| Zero $<= 2$ | 99.07% | 86.22% |

Table 1: Test accuracies of discriminative classifiers on MNIST, FashionMNIST test sets with different perturbations.

**Difference to Adversarial Examples**  Both the SITs we use in this paper and adversarial perturbations are small perturbations, but they are inherently different. Adversarial perturbations are intentionally crafted to cause misbehaviors, which are usually specific to individual images, and easily fool a classifier with almost $0\%$ accuracy. While three SITs above are universal transformations over all images that take no additional computations and basically come for free.

## 5 DISCUSSIONS AND CONCLUSIONS

**What is the problem of likelihood-based generative models?**  Discriminative classifiers, trained to extract class-relevant features, are known to be vulnerable to adversarial examples, and give over-confident predictions even for OoD samples. Generative models are supposed to be more robust since they model every pixel information of an image. However, likelihood modeling in high-dimensional space can be hard and lead to counter-intuitive observations. It was observed that likelihood-based generative models can assign even higher likelihoods on OoD samples(Nalisnick et al., 2018; Choi & Jang, 2018). Nalisnick et al. (2018) observe this phenomenon on both flows and VAEs. They decompose the *change-of-variable* theorem and investigate the influences of different transformation layers, find that the phenomenon still exists regardless of whether the transformation is *volume-preserving* or not. Their second-order analysis on pixel statistics suggests that OoD datasets, e.g. MNIST, just sit inside of in-distribution datasets, e.g. FashinMNIST, with roughly the same mean, smaller variance. They suspect that flows may simply fit the pixel intensities without really capture the high-level semantics. Ren (2019) find that the likelihood of an image is mostly dominated by the irrelevant background pixels, and propose a remedy to correct the original likelihood with a likelihood ratio. Though significantly improves the accuracy of OoD detection, but still fail to answer the question: whether the likelihood ratio shows high correlation to high-level semantics.

This paper differs from previous works and step further to explore the correlations between the likelihood of flow-based generative models and image semantics. Theoretical analyses in (Theis et al., 2015; van den Oord & Dambre, 2015) point out an important argument that generative models' ability to produce plausible samples is neither *sufficient* nor *necessary* for high likelihood. Results in this paper provide more experimental evidences for this simple argument that even for powerful exact likelihood-based generative models-flows, the likelihoods of samples can be largely weakly

correlated to the high-level image semantics. Thus, special attention should be paid to this argument before we apply likelihood-based generative models to downstream tasks. For example, considering the weak correlation between flows' likelihoods and image semantics, it may be inappropriate to use them for OoD samples detection. On the other hand, these counter-intuitive behaviours of flows raise our awareness of the gap between the predictive likelihoods of flows and the expectation that these likelihoods can closely relate to the semantics for OoD detection.

**What is exactly the *likelihood* of a image?** We should keep in mind that the predictive likelihood of a flow is the joint probability of all the image pixels. There is no doubt that flows, trained by maximizing its likelihood, could generate impressive synthesized data. There seem to be no problem that in terms of image generation, we expect that every single generated pixel in a image is the most likely one (hinging on its contextual pixels). However, the likelihood is explicitly modeled on pixels, so can be easily influenced by pixel-level modifications. Images' likelihoods significantly decrease even small noises are added to the pixels of backgrounds. For downstream tasks that need some "likelihood" to indicate the object in an image is a cat, rather than a car, the pixels of backgrounds are almost irrelevant. This drive us to think that we may need to model likelihood in some kind of *semantic* space or with some "perceptual" metrics, rather than on raw pixels. One promising direction is to define likelihood of an images on its high-level representation, and successful examples are (Lee, 2018; Nilesh A. Ahuja, 2019).

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

## A  SETUP OF EXPERIMENTS AND MODELS

The target models Glow and PixelCNN are implemented in Pytorch 1.0, and the implementation of PixelCNN is based on **https://github.com/pclucas14/pixel-cnn-pp**. Though several modifications proposed in PixenCNN++ are used in our implementation, we still use PixelCNN to annotate our model. We implement both unconditional and conditional versions of Glow. We report the results of conditional Glow in our experiments.

For Glow, using multi-scale architecture with more levels is critical to achieve lower BPDs. We resize the FashionMNIST, MNIST images from $28 \times 28$ to $32 \times 32$ and set the number of levels to 5. So the last factored the latent variables will be $1 \times 1$ in the spatial size. We provide pretrained models in code package as supplementary material for reproducing the results (We only provide models on MNIST due to the upload size limit). For PixelCNN, both $28 \times 28$ and 32 versions are provided.

In image translation experiments, we report the BPDs of PixelCNN on MNIST images of size $28 \times 28$ in the paper. Unlike Glow, PixelCNN treat the pixels as discretized intensity levels, and resizing the image size to $32 \times 32$ could lead to dangling intensity levels. We also find that if performing image translation experiment on PixelCNN($32 \times 32$), 1-pixel or 2-pixel translation will also lead to considerably BPD decreases.

Table 2: Hyperparameters of Glow

| Key | Value |
| --- | --- |
| Optimizer | Adam |
| learning rate | 0.0002 |
| batch size | 64 |
| width of NN module | 128 |
| depth (# step flow blocks) | 8 |
| n_level (# scale levels) | 5 |

Table 3: Hyperparameters of PixelCNN

| Key | Value |
| --- | --- |
| Optimizer | Adam |
| learning rate | 0.0002 |
| batch size | 50 |
| # mixture components of logistic | 3 |
| # filters | 100 |
| # residual blocks per stage | 2 |

# B ADDITIONAL RESULTS OF IMAGE TRANSLATION

## B.1 ADDITIONAL RESULTS

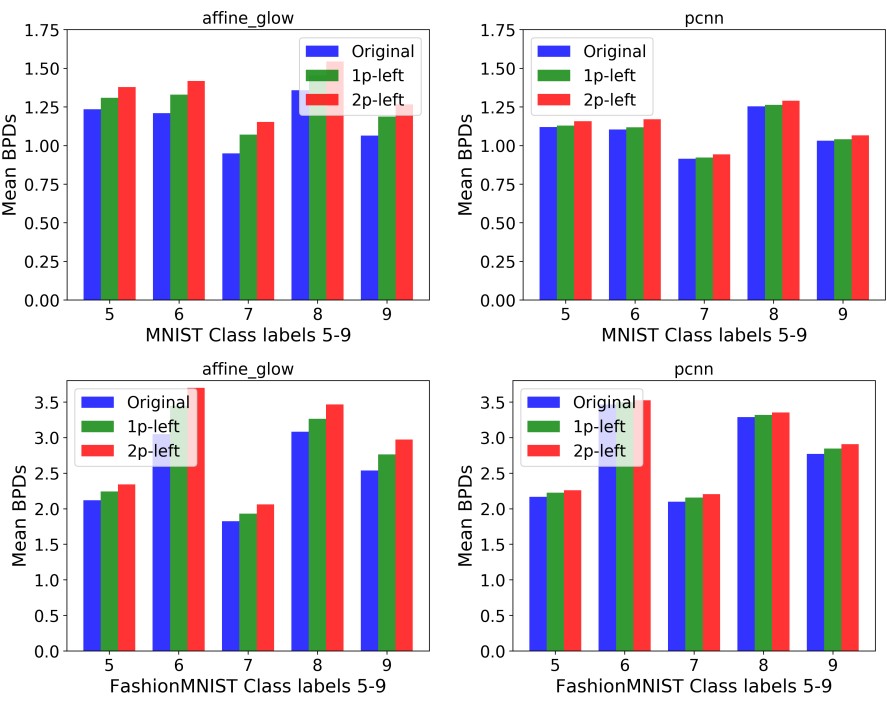

Figure 6: BPDs of (affine) Glow and PixelCNN on MNIST, FashionMNIST test sets with different 1-pixel and 2-pixel left translation (1p-left, 2p-left). Shown are classes from 5-9.

## B.2 RESULTS OF ADDITIVE GLOW

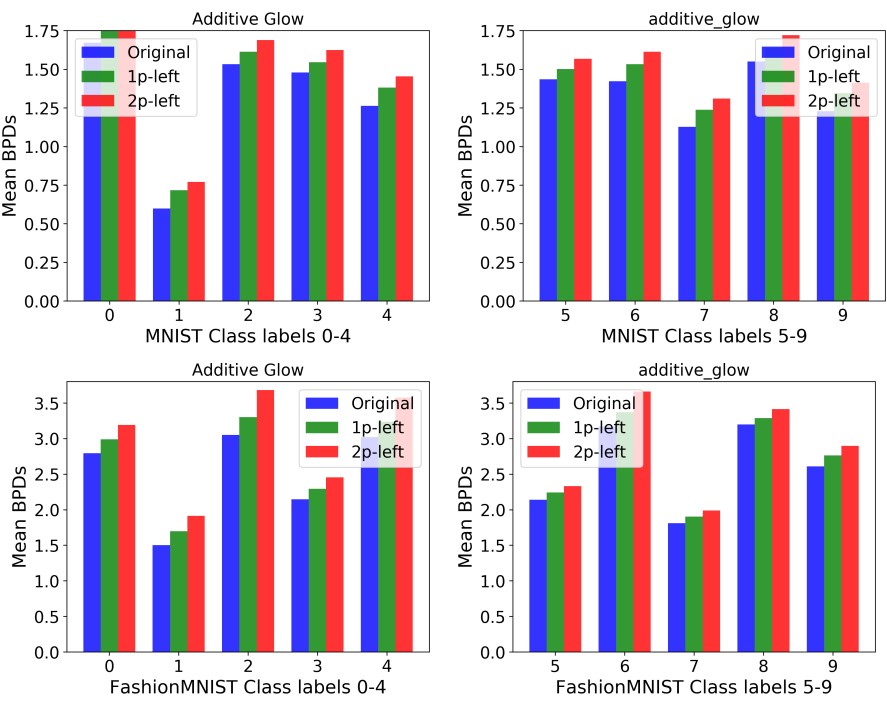

Figure 7: BPDs of (additive) Glow on MNIST and FashionMNIST test sets with different 1-pixel and 2-pixel left translation (1p-left, 2p-left).

## C  ADDITIONAL RESULTS OF ZEROING LATENT FACTORS OF GLOW

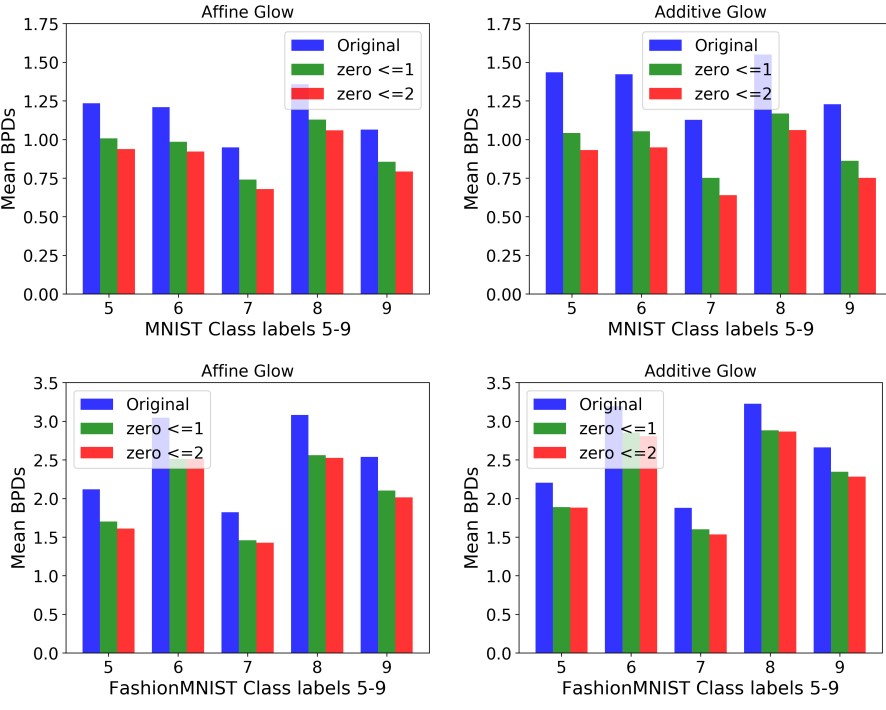

Figure 8: BPDs of affine and additive Glow on MNIST, FashionMNIST test sets by zeroing the preceding 1 (zero <=1) and 2 (zero <=2) latent factors.

## C.1 Examples of Additive Glow

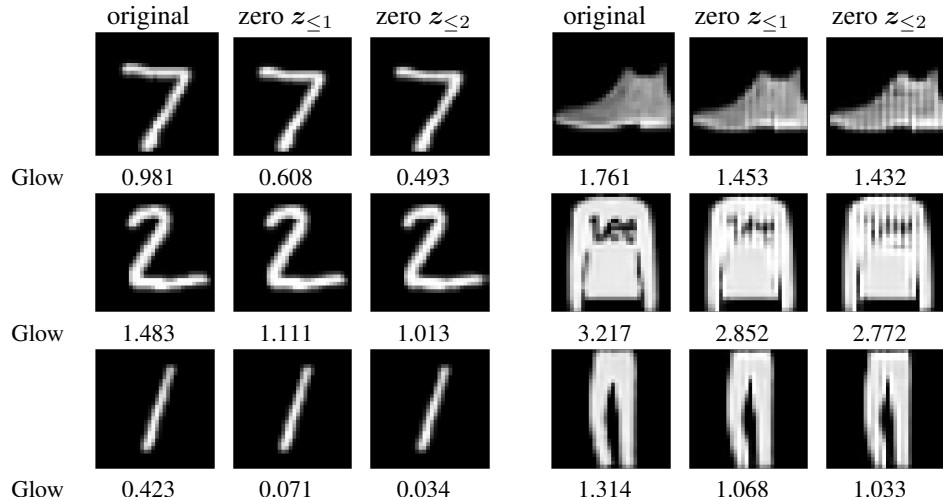

Figure 9: Examples of Additive Glow with different zeroed latent variables on MNIST (left) and FashionMNIST (right) test sets.

## D    ADDITIONAL EXAMPLES OF RANDOM NOISE PERTURBATION

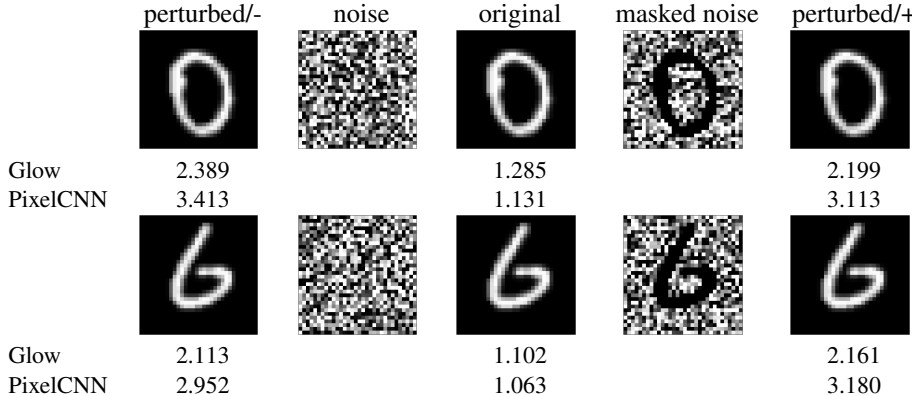

Figure 10: Examples from MNIST test sets perturbed with gaissian noises. The middle column shows the original images. The second and fourth columns are the gaussian noises and masked gaussian noises. The first and fifth columns are the perturbed images with noises w/o (+/-) masks. Below the samples are the BPDs reported by Affine Glow and PixelCNN.

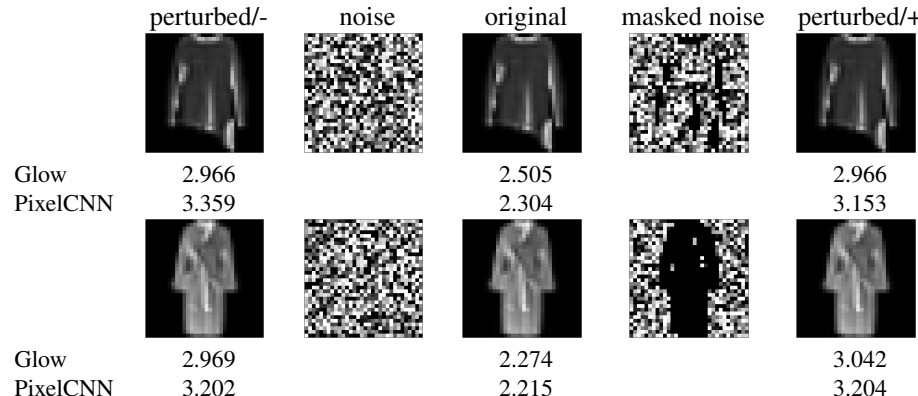

Figure 11: Examples from FashionMNIST test sets perturbed with gaissian noises. The middle column shows the original images. The second and fourth columns are the gaussian noises and masked gaussian noises. The first and fifth columns are the perturbed images with noises w/o (+/-) masks. Below the samples are the BPDs reported by Affine Glow and PixelCNN.

