# OpenReview forum: "Exploring the Correlation between Likelihood of Flow-based Generative Models and Image Semantics"
_ICLR.cc/2020/Conference — Reject_

### Official Review · AnonReviewer1 · 2019-10-16
**Official Blind Review #1**

**Rating:** 1

**Review:**

Summary:

The paper studies likelihood-based models of images, such as Glow and PixelCNN. The paper shows empirically that image transformations that preserve semantics (e.g. translations by a few pixels) produce images that have lower probability (density) under such models.

Decision:

The paper is studying an important topic, which is how to use likelihood-based models correctly, and it warns against the misuse of such models. I agree with the paper's conclusion that we should be careful when using likelihood-based models.

Nonetheless, in my opinion the paper is a clear reject. The paper is full of flaws, incorrect statements, poorly constructed arguments, speculative explanations, and superficial descriptions of previous work.

Broadly, the main issue with the paper is the following:
(a) It begins with flawed assumptions about how a likelihood-based model is expected to behave.
(b) It tests two likelihood-based models experimentally and finds that they don't behave according to the assumptions.
(c) It concludes that we need to be cautious when using likelihood-based models.

As I said, I agree that we should be careful when using likelihood-based models, but I worry that the way the paper reaches this conclusion can mislead and misinform readers. In what follows, I elaborate on specific issues with the paper in more detail.

Issue #1:

The main flaw of the paper is the assumption that semantic-preserving transformations shouldn't reduce the likelihood of the model (beginning of section 4). This is incorrect. To see why, consider a semantic-preserving transformation x' = T(x). As defined in the paper, a semantic-preserving transformation is one that doesn't change the label y of an image x. This can be formalized as:

p(y | x') = p(y | x)

By Bayes' rule, from the above it follows that:

p(x' | y) p(y) / p(x') = p(x | y) p(y) / p(x)
=> p(x' | y) / p(x') = p(x | y) / p(x)

Clearly p(x') can be different from p(x), as long as p(x' | y) is different from p(x | y) by the same factor. Hence, it doesn't follow that if p(y | x') = p(y | x) then p(x') = p(x), which is what the paper incorrectly assumes. To be clear, in the above expressions, p() refers to the true data distribution and not to a model that approximates it.

For example, consider images of digits, where the digit is generally in the centre of the image. Moving a digit to the corner will result in a less likely image, because it's unlikely that digits appear in corners. However, it won't change the classification of that digit, since all digits are less likely to appear in corners in exactly the same way. In fact, this is exactly why we see in section 4.1 that the likelihood of the model decreases as the image is translated to the left; the model is behaving exactly as it is supposed to.

Similarly, in section 4.2 where noise is added to the image, the model is again behaving exactly as it is supposed to. Adding Gaussian noise to the image results in a distribution p(x') that is equal to convolving the original distribution p(x) with the noise distribution p(noise) which is an isotropic Gaussian. As a result, p(x') will be a more diffuse version of p(x), hence samples x' will have on average low probability (density) under p(x), exactly as expected, and exactly as the experiment observes.

Issue #2:

The paper incorrectly assumes that out-of-distribution examples should have low probability (density). This is incorrect, and a common misconception that results from confusing high probability with typicality. In fact, out-of-distribution examples can have high probability (density). Here are two examples that illustrate that:

Suppose you flip a bent coin a million times, with 10% probability of the coin coming up heads. The in-distribution samples (the typical set) are those sequences of coin tosses that have roughly 100 thousand heads. However, the most likely outcome is the all-tails sequence. This outcome is clearly atypical, and many people would agree that it's out-of-distribution, but it has the highest probability.

Consider one million independent Gaussian variables, each with mean 0 and variance 1. Due to the law of large numbers, a typical draw of these variables will have average squared value very close to 1, hence the outcome of all variables being zero is very atypical and many people would agree it's out-of-distribution. However, the all-zero outcome is in fact the one with the highest probability density.

Given the above, the following two statements copied from the paper are flawed, and potentially misleading:

"The foundation of using likelihood-based models for OoD detection is that they are supposed to assign much lower likelihoods for OoD samples than in-distribution samples"
"In OoD detection, we assume that a sample with a higher likelihood indicates that it is more likely to be an in-distribution sample"

Fundamentally, I think the issue is that the paper incorrectly assumes that all images with the same semantics (e.g. all images of the digit 3) must be in-distribution. However this is not necessarily true. For example, the true data-generating process of MNIST images (i.e. asking people to write down a digit, scanning it, denoising it, cropping it and centring it) is unlikely to produce images where the digit is not in the centre or the background is noisy. Hence, the images considered in sections 4.1 and 4.2 are indeed out-of-distribution with respect to the true data distribution of MNIST, and are not adversarial examples of the models as the paper suggests.

Further issues:

The paper is ostensibly about flow models, but in fact very little is specific to flow models, and most of the discussion, where correct, applies to likelihood-based models in general. In fact, PixelCNN is not a flow model, even though the paper misleadingly describes it as such. PixelCNN can be used to model discrete random variables, whereas flow models are used for continuous random variables (flows for discrete random variables exist, but they are different from PixelCNN). That said, if PixelCNN is used to model continuous random variables then by reparameterization it can be viewed as a flow model with one layer, but that would be an unusual way to present it. Same for WaveNet.

"It is also believed that flows can be used to detect out-of-distribution(OoD) samples by assigning low likelihoods on them."
Believed by whom? A citation is needed here.

"Flows can roughly be divided into two categories [coupling flows and autoregressive flows]"
There are several flows that fall in neither of these categories, such as linear flows, residual flows, planar flows, radial flows, Sylvester flows, neural ODEs, FFJORD, and many others.

"The autoregressive property of an autoregressive layer is enforced by masking."
There are other ways of enforcing the autoregressive property (e.g. RNNs); masking is just one of them.

Eq. (5) is not correct in general. The bits per dimension should be approximated by:

BPD = (NLL - log|B|) / ((h x w x c) * log2)

where |B| is the quantization volume, provided |B| is small. That is, if each pixel with range [0, 1] is quantized into 10 bins, then |B| = 0.1 ^ (h x w x c).

"This surprising difference can be attributed to the difference of their architectures" (beginning of page 5)
The explanation that follows is speculative, but is not presented as such, which can be misleading.

"We may reasonably suspect that flows’ counter-intuitive likelihood assignment is dominated by the inherent differences of pixel-level statistics associated to the image semantics"
This is also speculation.

"PixelCNN is more sensitive to the noises, because its pixel-wise modeling quickly augment and propagate the influences of the added noise"
Also speculation.

"We find that the semantic object of a test image depends heavily on the last factored latent zL, rather than the preceding factors"
As far as I can see, there isn't evidence in support of that statement in the paper.

"Considering the weak correlation between flows’ likelihoods and image semantics, it is inappropriate to use them for OoD samples detection"
Given the flawed assumption about the role of image semantics, I don't think there is evidence for that.

"In terms of image generation, we expect that every single generated pixel in a image is the most likely one"
This is inaccurate; when generating images from a model, we don't get the most likely pixels, but samples from the joint distribution over pixels.

**Experience Assessment:**

I have published in this field for several years.

**Review Assessment: Checking Correctness Of Derivations And Theory:**

I carefully checked the derivations and theory.

**Review Assessment: Checking Correctness Of Experiments:**

I carefully checked the experiments.

**Review Assessment: Thoroughness In Paper Reading:**

I read the paper thoroughly.

---

> ### Author Response · Authors · 2019-11-09
> **Responses (part 1/2) [You made a factual mistake about out-of-distribution, which is critical]**
>
> We are grateful for your detailed and instructive reviews.
>
> Before we start make responses to the issuses you raised, I have to (informally) define two types of likelihood-based models:
>
> Type I: Exactly the current flow-based models (e.g. PixelCNN, Glow or many other variants), modeling precisely the likelihood of an image $x$ with the joint product of all the pixels of $x$.
>
> Type II: Models whose likelihoods are robust, and fit humans' intuitions, e.g. naturally assigning lower likelihoods for out-of-distribution samples  (this is critical for potential  likelihood-based models to be applicable to downstream tasks like OOD detection. This is also something publicly recognised , see abstracts of [1] [2]).
>
>  Your statement that our paper is "full of flaws, incorrect statements,  ... ", is probably (we think) because you limit the likelihood-based models to Type I models, and implicitly rule out any possibility of the existence of Type II models (examples of Type II are provided at the bottom).
>
> Issue 1:
>
> You say that our observations (pixel shifts, adding Gaussian noises) lower the likelihoods of samples are actually expected. I completely agree. But note this applied only to Type I models.
>
> This paper is NOT to show  some surprising (actually expected) observations of Type I models, BUT to show the gap between Type I (what we got) and Type II models (what we expect) in terms of OOD detection.
>
> Issue 2:
>
> The typicality you introduced and OoD are \emph{unrelated} concepts. Thus the two examples you proposed are irrelevant and misleading.
>
> There is a clear fact about in-distribution and OoD is that they are perceptually two \emph{different} distributions. Typical examples, proposed in [1] [2], are in-out distribution pairs: FashionMNIST(in)- MNIST(out), CIFAR10(in)-SVHN(out).
>
> In our examples, typical or atypical outcomes are all discussed  within the SAME distribution proposed. So there are no OoD samples at all.  The atypical samples have nothing to do with OoD samples.
>
> A simple example, for a generative likelihood model $g_{dog}$ trained on only dog images, then a cat image $x_{cat}$ is OoD sample (thus no way $g_{dog}$ is supposed to assign high likelihood to $x_{cat}$).
>
> A mistake in your "Gaussian variables" example: all-zero is the highest point of  pdf of the resulting Gaussian, so no way it is OoD.  If you are talking about the probability, all-zero is a point of for a continuous random variable, whose probability is exactly zero, so do any other points.  This explains the atypicality you raised.
>
> Please REconsider this statement: "Fundamentally, I think the issue is that the paper incorrectly assumes that all images with the same semantics (e.g. all images of the digit 3) must be in-distribution. However this is not necessarily true." This is very true in terms of OOD detection.
>
> And we show that semantic-invariant transformations (SITs) are inherently different from adversarial examples (AEs) in that: (1) AEs are specially crafted, while SITs are not; (2) AEs can reduce classifiers' accuracies to ~0%, while SITs hardly do so.
>
>
> [1] Nalisnick, Eric, et al. "Do deep generative models know what they don't know?." arXiv preprint arXiv:1810.09136 (2018).
>
> [2] Ren, Jie, et al. "Likelihood Ratios for Out-of-Distribution Detection." arXiv preprint arXiv:1906.02845 (2019).

---

> > ### Author Response · Authors · 2019-11-09
> > **Responses (part 2/2)**
> >
> > Further issues:
> >
> > "Flows can roughly be divided into two categories [coupling flows and autoregressive flows]"
> >
> > (1) This paper only focus on Flow-based generative models (their likelihood behaviours), not the general flow models. (2) We divide the flows based on the granularity, i.e. coupling (2 parts) or autoregressive (# pixel parts).  We will update and clarify.
> >
> > "We find that the semantic object of a test image depends heavily on the last factored latent zL, rather than the preceding factors"
> >
> > This is exactly supported by the examples provided. The image semantics keep unchanged, even if we zero the preceding factors, as long as we keep the last factored latent still)
> >
> > "PixelCNN is more sensitive to the noises, because its pixel-wise modeling quickly augment and propagate the influences of the added noise"
> > This is not speculation. PixelCNN is pixel-wise autoregressive (fine-grained), while Glow bases on coupling layers(coarse-grained).
> >
> > "In terms of image generation, we expect that every single generated pixel in a image is the most likely one"
> >
> > We mean here that very single likely generated pixel  hinges on the contextual pixels, i.e. the joint distribution over pixels.
> >
> > Will revise for clarification on these issues.
> >
> >
> > Final:
> >
> > You said "(a) It begins with flawed assumptions about how a likelihood-based model is expected to behave." which is not True.  Do you think it is a flawed assumption that generative model $g_{dog}$ should assign higher likelihood to a dog image $x_{dog}$ than a cat image $x_{cat}$?
> > Flows define likelihood as the joint probability of all image pixels.  However, is this the only way that the "likelihood" of an image should be defined, considering flows' counter-intuitive behaviours ?  A simple alternative way is to model the image likelihood on its high-level representations. For example, [3] models class conditionals on logits of a discriminative classifier, i.e. conditional likelihood, and performs very well on OOD detection. Another similar example is [4].
> >
> > All in all, this paper aims to show the gap between flows and likelihood-based models we expect on OoD detection.  We call for attention that the community should rethink what is exactly the \emph{likelihood} of an image except for joint probability of pixel-level intensities.
> >
> > [3] Lee, Kimin, et al.  "A simple unified framework for detecting out-of-distribution samples and adversarial attacks", NIPS 2018
> >
> > [4] Nilesh A. Ahuja, et al. "Probabilistic Modeling of Deep Features for Out-of-Distribution and Adversarial Detection", https://arxiv.org/abs/1909.11786

---

> > > ### Comment · AnonReviewer1 · 2019-11-15
> > > **Response**
> > >
> > > Thank you for your response.
> > >
> > > If I understand correctly, your main point is that the joint probability of pixels is not an appropriate quantity to use for certain downstream tasks. The reasons being: (a) it is not invariant to semantic-preserving transformations, and (b) it is not suitable for out-of-distribution detection.
> > >
> > > I completely agree with the above point. In fact, my review makes exactly the same point: issue #1 explains why (a) is the case, and issue #2 explains why (b) is the case.
> > >
> > > In addition, my review explains that (a) and (b) above are in fact properties of the true data distribution, and not properties of a particular model (flow or otherwise). A model that approximates the data distribution will naturally inherit properties (a) and (b). I think this is an important point that I don't think the paper makes clear. Instead, the paper focuses on flow-based models, and gives the impression that the problem lies with the model, whereas in reality the problem is that the data distribution itself doesn't have the properties that you desire (what you refer to as "type II").
> > >
> > > In the following, I'd like to clarify a few more points:
> > >
> > > The term "likelihood" has a very specific meaning in machine learning and statistics: it is the probability of the observed data as a function of the model parameters. See for example:
> > > - The Wikipedia article: https://en.wikipedia.org/wiki/Likelihood_function
> > > - Page 29 of MacKay's book: https://www.inference.org.uk/itprnn/book.pdf
> > > Therefore, it makes sense to talk about "the likelihood of a model" but it doesn't make sense to talk about "the likelihood of an image". Moreover, defining the likelihood to have potentially different meanings (like you did in your response above) may confuse some readers and make it hard for them to understand your point. If you'd like to discuss models that assign various types of "scores" to an image other than the joint probability of its pixels, then I suggest that you define precisely what you mean, and be careful with the terminology used.
> > >
> > > In "issue #2", I explained how atypical examples can have higher probability than typical examples. This is an established mathematical fact, and is not up for debate. The concept of a "typical set" has a precise mathematical definition, see for example:
> > > - Section 4.4 of MacKay's book: https://www.inference.org.uk/itprnn/book.pdf
> > > - Section 2 of Nalisnick et al.'s paper: https://arxiv.org/pdf/1906.02994.pdf
> > > If you'd like to see in full mathematical detail why the mean of a high-dimensional Gaussian is not in the typical set even though it has the highest density, please work through Exercise 6.14 in MacKay's book.
> > >
> > > The reason I brought up typicality in the "out-of-distribution" discussion is that because I believe that typicality is a suitable formalization of the concept of "out-of-distribution". I think there is a growing realization in the machine-learning community that the two are closely related, see for example Nalisnick et al.'s paper https://arxiv.org/pdf/1906.02994.pdf on this exact subject.
> > >
> > > If you believe that out-of-distribution and typicality are not related, then that's fine. Unlike typicality, "out-of-distribution" doesn't have a precise meaning, and is therefore up for interpretation. However, if you want to make precise statements about the notion of "out-of-distribution", I strongly suggest that you give it a precise mathematical meaning first. Talking about dogs and cats can be helpful and intuitive, but it doesn't formalize the concept.
> > >
> > > At the end of your response, you suggest modelling images in an alternative space, e.g. a higher-level representation instead of pixels. I think that's a good idea: I encourage you to pursue this direction, and I welcome more papers on this topic. As I said earlier, if the conclusion of the paper is that for many downstream tasks the joint probability of pixels isn't appropriate, then I agree.
> > >
> > > In conclusion, I think that the paper as is currently written is not ready for publication. The conclusion that the joint probability of pixels is not invariant to semantic-preserving transformations should be fairly obvious to people familiar with probabilistic modelling, and as reviewer 3 also pointed out, is not enough for a full paper. Moreover, there is serious misuse of terminology that can easily mislead and confuse readers (as evidenced by this entire discussion). Finally, the paper makes statements about imprecise notions (such as the concept of out-of-distribution) that don't hold up to scrutiny. In my opinion, a paper that warns about the misuse of likelihood-based models should be careful and precise when dealing with nuanced notions of probability theory, otherwise it may do more harm than good, especially for inexperienced readers. For these reasons, I will maintain my score, but I will also encourage the authors to reflect carefully on our discussion and take it into account when revising the paper in the future.

---

> > > > ### Author Response · Authors · 2019-11-15
> > > > **Responses**
> > > >
> > > > Thank you for your comments.
> > > >
> > > > Though you will maintain your score, I am still glad that all our discussions get much clear. I also basically agree with all the points in your responses. With only one exception about the precise mathematical definition of "out-of-distribution".
> > > >
> > > > By "out-of-distribution", I insist on that it is rooted in humans' perception about the high-level information e.g. semantics, which also fits the requirements of downstream applications. Also the typical classification task in ML-community aims also for high-level semantics recognition. So this almost make it impossible to give a precise mathematical definition. That is also why I think out-of-distribution $\neq$ typicality you introduced.
> > > >
> > > > I agree mathematical rigor is preferable. But I think it's hard for this. Do you have any further comments?

---

> > > > > ### Comment · AnonReviewer1 · 2019-11-15
> > > > > **Response**
> > > > >
> > > > > Thank you for your response.
> > > > >
> > > > > I'm glad we agree. I think this was a fruitful conversation overall, and I hope it will improve the paper going forward.
> > > > >
> > > > > To me it seems that by out-of-distribution you mean something like "the set of all images x for which P(y=dog | x) < epsilon". But in any case, it's good to be precise, mathematically or otherwise.
> > > > >
> > > > > I don't have any other comments. Thank you again for the discussion.

---

### Official Review · AnonReviewer2 · 2019-10-23
**Official Blind Review #2**

**Rating:** 3

**Review:**

This paper raises a problem of the robustness of (log) likelihood computed by invertible flows. The authors show that the changes of likelihood of an image computed by flow-based image generative models have surprisingly weak correlations with semantic changes of image. The flow likelihoods are sensitive to very small changes of pixels that do not affect the semantics of an image. And the likelihoods are less robust against out-of-distribution inputs where we expect strong robustness compared to discriminative models.
These claims are validated and supported by several simple experimental results.

This is an interesting paper: it warns the abuse of likelihoods computed by flow models with several numerical experiments, which are simple but clearly designed to support the claims.
These experimental results convince me that the likelihoods of flow-based image generative models are joint distributions of pixel intensities and it is natural such likelihoods are apt to be sensitive in pixel intensity changes, even if they are semantically meaningless.

Designs of experiments are apparently similar to those of (Nalisnick+, 2018) at the first glance.  I think there is a room to improve the manuscript to clarify the difference from the Nalisnick+’s work. My understanding is that (Nalinsnick+, 2018) is interested in the OOD likelihood behaviors when datasets are swapped, and explains the behaviors of OOK likelihood based on the variance and the curvature of the dataset. This paper directly manipulates the pixel intensities so small that the statistics of the images would not change.
BTW, the Nalisnick’s paper is accepted and published in ICLR 2019.

It is a well-known fact that the flow (glow) models can generate natural and high-resolution images by interpolating ``latent hidden vectors’’. This indicates the latent representations are robust against perturbations while pixel intensities are not. So, when this transition of robustness occurs? This seems an interesting problem for me. I’m happy to hear the authors’ opinions about this issue.

Summary
+ Good research question concerning the robustness of flow models
+ Simple and understandable claims supported by simple experiments
+ Easy to read
- Can be improved more to clarify the difference from the previous work that study the flow model’s likelihoods.


**Experience Assessment:**

I do not know much about this area.

**Review Assessment: Checking Correctness Of Derivations And Theory:**

I did not assess the derivations or theory.

**Review Assessment: Checking Correctness Of Experiments:**

I assessed the sensibility of the experiments.

**Review Assessment: Thoroughness In Paper Reading:**

I made a quick assessment of this paper.

---

> ### Author Response · Authors · 2019-11-10
> **Responses**
>
> We are grateful for your comments.
>
> I think what you point out boils down to the fundamental problem: how to evaluate generative models (GAN, VAE, and Flows, etc), which is still an open problem.
>
> Here, we focus on Flow-based generative models since they provide exact likelihood evaluation of samples. Let's do some quick Q&A:
>
> (1) Is likelihood a good measure for evaluation of generative models?
>
> The answer is NO.
>
> The first thing we care about generative models is the quality of the generated samples. We may implicitly think that good likelihood implies good generation quality, which theoretically is simply not True. In practice maximizing the likelihood is always a  way, but hardly the best way,  to improve the samples' quality.  (Maximizing the data likelihood is equivalent to minimizing the KL-divergence between real data distribution $p_{data}$ and our model distribution $p_{model}$. KL is not a good measure, but is computationally possible or affordable. [1] also discusses the differences of optimizing different measures, e.g. MMD, JSD.)
>
> [1] has made a very clear argument about this by theoretical analyses, (we also mentioned and discussed in this paper):
>
> "Good likelihood is neither necessary nor sufficient to good generation quality (i.e. plausibility of samples)"
>
> The observations in this paper provide more experimental evidences for this simple but important argument.
>
> Then get back to Glow. There is no doubt that simple interpolations in the latent spaces of Glow still give impressive (semanticaly meaningful) high-resolution images. But it is not clear how the quatitative likelihood values of thses interpolations vary, up or down? Following the procedures proposed in this paper, we may still get high-quality images with much lower likelihoods (pixel-shifts or adding small noises) or much higher likelihoods (zeroing preceding latents).
>
> (2) Is there an universal measure for evaluation of generative models? (Or should we aim to find the universal one?).
>
> The answer is probably not.
>
> Evaluation of generative models depends heavily on different uses of them. This is also pointed out by [1] and also mentioned in Ian Goodfellow's Deep Learning Book [2] (section 20.14, page 717-719).
>
> A typical example is exactly Glow. If we only care about quality of generated images, it seems there is nothing wrong about Glow. However when it comes to deploy Glow for OOD detection, we see counter-intuitive behaviours.  If we only focus on  the  quantitative likelihood values of flows without asking or exploring the messages behind,  we may get in trouble  when they are deployed on downstream tasks.
>
>
> We will include discussion of Nalisnick’s paper in our revisions.
>
> [1] [A note on the evaluation of generative models](https://arxiv.org/pdf/1511.01844.pdf) by Theis, Lucas and Oord, Aaron van den and Bethge, Matthias.
>
> [2] Deep Learning, Ian Goodfellow and Yoshua Bengio and Aaron Courville.

---

### Official Review · AnonReviewer3 · 2019-10-23
**Official Blind Review #3**

**Rating:** 3

**Review:**

The paper studies the correlation between likelihood of flow-based generative models and image semantic information, and shows that even small perturbations, like a few pixel translations or noise applied to background, significantly affect models’ likelihoods, which signals that these likelihood models cannot be used for out-of-distribution data detection. However, very similar observations were made in prior works [1] and [2]. In particular, the paper [2] showed that likelihood of PixelCNN is dominated by background pixels which makes the observations in section 4.2 (applying noise to background) unsurprising. The sensitivity of Glow model to even 1-2 pixel translations (section 4.1) and exploiting multi-scale structure of Glow (zeroing latent variables in section 4.3) are interesting, but I believe, not enough for a full paper. Thus, due to the limited novelty, I recommend a weak reject.

Other questions and concerns:
1. The author claim to introduce “semantic-invariant transformation”. I believe this can be called “data augmentation”, why introduce a new term?
2. The last bullet point in the introduction is not clearly written.
3. Equation 1: variable u wasn’t introduced. Paragraph after equation 4: please fix the comma.
4. The clarity of figure / table captions can be improved, as well as their references in the main text.
5. The section 4.4 is confusing. Which discriminative classifiers are considered? How are they trained? The Table 1 is not referenced in the main text and the results are not explained or discussed.
6. The experiments are only performed on MNIST / FashionMNIST datasets. It would help to see experiments on other datasets, e.g. CIFAR-10, SVHN.
7. Related work section can be elaborated: please, discuss how the observations made in the paper are different from / consistent with [1] and [2].


[1] Nalisnick, Eric, et al. "Do deep generative models know what they don't know?." arXiv preprint arXiv:1810.09136 (2018).
[2] Ren, Jie, et al. "Likelihood Ratios for Out-of-Distribution Detection." arXiv preprint arXiv:1906.02845 (2019).


**Experience Assessment:**

I have read many papers in this area.

**Review Assessment: Checking Correctness Of Derivations And Theory:**

N/A

**Review Assessment: Checking Correctness Of Experiments:**

I carefully checked the experiments.

**Review Assessment: Thoroughness In Paper Reading:**

I read the paper thoroughly.

---

> ### Author Response · Authors · 2019-11-09
> **Responses**
>
> We are grateful for your comments.
>
>
> This paper is not to reintroduce the observations already made in [1] [2]. And our work is based on their observations, I agree that we should include more discussions about this paper and [1] [2].
>
> This paper is not to show that some "surprising" observations we made about flows. You point out: (1) the observation likelihoods of flows can be influenced by adding noises or pixel-shifts are unsurprising, which I completely agree, and I also think that trivial observations like these should not be accepted by ICLR; (2) observations made about multi-scale architecture of Glow are interesting, but not enough for a full paper, which I also agree.
>
> However, this paper is: (1) to demonstrate that what is normal and unsurprising observations for flows can be problematic when applied to downstream tasks like OOD detection; (2) to call for attention the gap between what likelihoods of flows actually are and what we expect likelihood-based models to be.  Our experiments and analyses motivate that we may not restrict the likelihood of an image to be a pixel-level definition we have now, i.e. the joint probability of all pixels. We should explore more robust likelihood-based models, i.e. fit humans' intuitions, e.g. naturally assign low likelihoods to out-of-distribution samples.  For example, we may model the likelihood of an image on its high-level representations, so that the likelihoods could have higher correlations to the high-level information of image, e.g. image semantics. (See also our responses to reviewer 1 for  more details. )
>
> Research on Flow-based models is a rapidly evolving field. But most works focus on the design of the bijective transformation layers to achieve lower bits-per-dim (BPDs) or equivalently higher likelihoods on standard datasets. Much less attention has been paid to explore the behaviours and properties of the likelihoods of flows reported, as well as their applicabilities to the downstream tasks like OOD detection.  So you may have underestimated the messages we want to send to the community.
>
> Other questions and concerns:
>
> 1. We introduce "semantic-invariant transformation" deliberately to distinguish the transformations used in this paper from the concept "data augmentation". "Data augmentation", from our perspective, aims to improve the generality of classifiers at inference, which are quite different from what we do here. We don't do this to earn inappropriate credit.
>
> 2.3.4 We will revise.
>
> 5. We will add details of the discriminative classifiers, and improve.
>
> 6. We'd like to do that if we have enough time.
>
> 7. We will add more discussions.
>
>
>
> [1] Nalisnick, Eric, et al. "Do deep generative models know what they don't know?." arXiv preprint arXiv:1810.09136 (2018).
>
> [2] Ren, Jie, et al. "Likelihood Ratios for Out-of-Distribution Detection." arXiv preprint arXiv:1906.02845 (2019).

---

### Author Response · Authors · 2019-11-15
**Revision uploaded based on all reviews, please check**

Thank you all reviewers for your time.  I really appreciate your efforts.

To reviewer 1:
Though some statements in your review looks a little bit hash, I find it is very helpful for me get a different picture of this topic, and make a clarification. Please do check my responses.

Thank you for pointing out some inaccurate statements in this paper.

To reviewer 2&3:

Discussions with related works you proposed are added in Section 5 Discussions and Conclusions.

To all:

Some minor issues are updated accordingly, please check the revision.

---

### Decision · Program_Chairs · 2019-12-19

**Decision:**

Reject

**Comment:**

This paper discusses the (lack of) correlation between the image semantics and the likelihood assigned by flow-based models, and implications for out-of-distribution (OOD) detection.

The reviewers raised several important questions:
1) precise definition of OOD: definition of semantics vs typicality (cf. definition in Nalisnick et al. 2019 pointed by R1)
There was a nice discussion between authors and the reviewers. At a high level, there was some agreement in the end, but lack of precise definition may cause confusion. I think adding a precise definition will add more clarity and improve the paper.

2) novelty: similar observations have been made in earlier papers cf. Nalisnick et al. 2018. R3 also pointed a recent paper by Ren et al. 2019 which showed that likelihood can be dominated by background pixels. Older work has shown that the likelihood and sample quality are not necessarily correlated. The reviewers appreciate that this paper provides additional evidence, but weren't convinced that the new observations in this paper qualified for a full paper.

3) experiments on more datasets

Overall, while this paper explores an interesting direction, it's not ready for publication as is. I encourage the authors to revise the paper based on the feedback and submit to a different venue.